# Cellular and Molecular Variations in Male and Female Murine Skeletal Muscle after Long-Term Feeding with a High-Fat Diet

**DOI:** 10.3390/ijms23179547

**Published:** 2022-08-23

**Authors:** Bright Starling Emerald, Mohammed A. Al Jailani, Marwa F. Ibrahim, Challagandla Anil Kumar, Mohammed Z. Allouh

**Affiliations:** Department of Anatomy, College of Medicine and Health Sciences, United Arab Emirates University, Al Ain 15551, United Arab Emirates

**Keywords:** satellite cells, myonuclei, myosin, fiber type, Pax7, Myh, myostatin

## Abstract

Current information regarding the effects of a high-fat diet (HFD) on skeletal muscle is contradictory. This study aimed to investigate the effects of a long-term HFD on skeletal muscle in male and female mice at the morphological, cellular, and molecular levels. Adult mice of the C57BL/6 strain were fed standard chow or an HFD for 20 weeks. The tibialis anterior muscles were dissected, weighed, and processed for cellular and molecular analyses. Immunocytochemical and morphometric techniques were applied to quantify fiber size, satellite cells (SCs), and myonuclei. Additionally, PCR array and RT-qPCR tests were performed to determine the expression levels of key muscle genes. Muscles from HFD mice showed decreases in weight, SCs, and myonuclei, consistent with the atrophic phenotype. This atrophy was associated with a decrease in the percentage of oxidative fibers within the muscle. These findings were further confirmed by molecular analyses that showed significant reductions in the expression of Pax7, Myh1, and Myh2 genes and increased Mstn gene expression. Male and female mice showed similar trends in response to HFD-induced obesity. These findings indicate that the long-term effects of obesity on skeletal muscle resemble those of age-related sarcopenia.

## 1. Introduction

Fatty diets have become a major part of daily nutrition. Consumption of a high-fat diet (HFD) for a significant period of time enhances the development of disease states, such as obesity and insulin resistance [1,2]. The substantial increase in HFD consumption over the past few decades has raised concerns over increasing morbidity and mortality associated with an HFD. An HFD increases the amount of lipid stored in various body tissues and subsequently amplifies the need for catabolism of these lipids. One of the most important tissues involved in this process is skeletal muscle.

Skeletal muscle, a major body tissue affected by HFD consumption, comprises thousands of elongated multinucleated cells known as “muscle fibers” [3]. The nuclei of these fibers do not replicate their DNA, and thus have limited regenerative capacity. The regenerative capacity of skeletal muscle is dependent on the presence of a specific population of adult myogenic stem cells, known as satellite cells (SCs) [4,5]. These cells are easily distinguished by their characteristic location between the sarcolemma and basal lamina of the muscle fiber and by their specific expression of the paired box transcription factor 7 (Pax7) protein [3,4,5,6]. When required for muscle growth or repair, SCs become active, proliferate, and finally differentiate by fusing with pre-existing muscle fibers or each other to form new fibers. At the end of this process, SC nuclei become new myonuclei [3,6].

In mammals, there are four major types of skeletal muscle fibers [7,8]. These are (i) slow-twitch oxidative type I fibers, (ii) intermediate-twitch oxidative type IIA fibers, (iii) fast-twitch oxidative glycolytic type IIX fibers, and (iv) fast-twitch glycolytic type IIB fibers [7,8]. Glycolytic type IIB fibers are only used to produce high force for a short period (phasic activity), because they fatigue rapidly. These fibers have a larger diameter and more glycogen reserves than oxidative fibers [3]. In contrast, oxidative fibers show higher fatigue resistance and are thus the main fibers used in routine continuous activities that require constant contraction for long periods, such as posture maintenance and walking [3].

HFD feeding affects skeletal muscle fibers at both structural and functional levels. At the functional level, HFD can induce metabolic alterations in muscle fibers, including impairment of oxidative capacity and increased insulin resistance [9,10,11]. However, at the structural level, there have been several contradictory reports on the influence of HFD on the morphology and composition of skeletal muscle fibers.

It was speculated that HFD would increase the cross-sectional area (CSA) of the muscle fibers due to the accumulation of intramyocellular lipids and interruption of their catabolism [12]. Shortreed et al. reported that an HFD increased the CSA of skeletal muscle fibers without increasing the levels of intramyocellular lipids [10]. Another study reported that an HFD augmented muscular atrophy and reduced the CSA of skeletal muscle fibers in middle-aged mice [13]. Furthermore, it has been reported that an HFD enhances the transformation of fiber composition from glycolytic to oxidative [14]. However, other studies have reported the opposite result. Hua et al. indicated that an HFD induces a shift from type I to type II muscle fibers [15]. A study on the human rectus abdominis muscle found that obese humans have a lower percentage of type I oxidative fibers in this muscle [16]. Additionally, Kriketos et al. reported that the percent body fat in Pima Indians correlated negatively with the percentage of type I oxidative fibers and positively with type IIB glycolytic fibers [17].

This study aimed to clarify the aforementioned debates in the literature by investigating the effects of an HFD on different morphological, cellular, and molecular parameters of skeletal muscle. First, we examined the influence of an HFD on skeletal muscle fiber type, size, and composition. Second, we examined the impact of an HFD on the cellular distribution of myogenic satellite cells and muscle fiber myonuclei, as these are the main orchestrators of skeletal muscle growth and repair. Finally, we tested the influence of an HFD on the expression of 84 important genes involved in basic skeletal muscle function, development, and disease.

## 2. Results

### 2.1. Body and Muscle Weights

There were significant (*p* < 0.05) increases in the body weights of HFD animals of both sexes compared to control animals. In males, the increase in body weight due to HFD feeding appeared significant (*p* = 0.037) by the eighth week (Figure 1A), whereas the increase appeared significant (*p* = 0.039) in females by the fourth week (Figure 1B). When dissected, HFD animals showed a prominent accumulation of visceral fat (Figure 2).

However, the weight of the TA muscle decreased significantly (*p* = 0.015) in the HFD groups of both sexes compared to that in the control groups (Figure 3). In males, the mean TA weights in the control and HFD groups were 53.7 ± 3.1 and 46.3 ± 0.6 mg, respectively. In females, the mean TA weights in the control and HFD groups were 46.7 ± 2.5 and 39.7 ± 2.1 mg, respectively.

### 2.2. Fiber Morphology (Type, Size, and Intracellular Lipids)

Figure 4 shows the immunofluorescence labeling of different fiber types within the TA muscle of control and HFD mice. Table 1 and Table 2 show the variations between the control and HFD groups regarding the frequency and size of different fiber types within the TA muscle of both male and female mice, respectively. We observed no type I slow oxidative fibers in the TA muscles of male or female mice (Figure 4A, Table 1 and Table 2). In contrast, type IIB fast glycolytic fibers were the most common fiber type in the TA of both sexes (Figure 4D, Table 1 and Table 2).

Our findings revealed a significant (*p* < 0.05) decrease in the frequency of oxidative fibers (type IIA, *p* = 0.022 in females and both IIA, *p* = 0.047, and IIX, *p* = 0.030 in males) in the HFD groups compared with their controls. Conversely, we observed an increase in the frequency of glycolytic fibers (IIB) in the HFD groups; however, this increase was not statistically significant (females *p* = 0.155; males *p* = 0.092).

Additionally, there was a significant (*p* < 0.05) increase in the size of oxidative fibers in the HFD group compared to the control group in both sexes. In males, the size increase included both type IIA (*p* = 0.004) and IIX (*p* = 0.035) fibers (Table 1), whereas in females, it included only type IIX (*p* = 0.020) fibers (Table 2). There was no difference in the size of glycolytic fibers between the HFD and control groups of both sexes (Table 1 and Table 2). 

BODIPY intracellular lipid staining was performed to investigate the accumulation of intramyocellular lipids. Imaging of the lipid-stained fibers clearly showed an accumulation of a notable amount of intramyocellular lipid droplets in the HFD fibers compared to control fibers, and the formation of prominent sub-sarcolemmal lipid plaques in the HFD fibers (Figure 5).

### 2.3. Satellite Cells and Myonuclei

Immunofluorescence sections labeled for SC nuclei and MN from the TA muscles of the control and HFD groups are shown in Figure 6. No significant (*p* > 0.05) difference was observed in the mean length of SC nuclei between the control and HFD groups of either sex (male *p* = 0.211, female *p* = 0.065) (Table 3). All SC indices (SC frequency, SCs/mm, and SC concentration) showed significant (*p* < 0.01) differences between the control and HFD groups in both sexes. The frequency of SCs was significantly (male *p* ≤ 0.001, female *p* ≤ 0.001) reduced in all HFD mice compared to control mice, as was the mean number of SCs per fiber unit length (1 mm) (male *p* ≤ 0.001, female *p* ≤ 0.001). The surface area of the sarcolemma per SC was significantly (male *p* ≤ 0.001, female *p* ≤ 0.001) greater in the HFD groups than in the control groups of both sexes, which indicated that SCs were more distant from each other (less concentrated) in the HFD animals than in the control animals (Table 3).

No significant (male *p* = 0.721, female *p* = 0.336) difference in the length of MN was observed between HFD and control animals of either sex (Table 3). The mean number of myonuclei per fiber unit length (1 mm) was significantly (male *p* = 0.003, female *p* = 0.004) lower in all HFD mice compared to that of control mice (Table 3), indicating that long-term feeding with an HFD induced myonuclear degeneration in TA muscle fibers.

The size of the myonuclear domain (MD), which is defined as the volume of sarcoplasm per myonucleus, was calculated for the TA muscles of the control and HFD groups of both sexes. There was a significant (male *p* = 0.006, female *p* = 0.002) increase in MD size in TA muscles of HFD mice compared to control mice of both sexes (Table 3). The MD increase in HFD mice could be attributed to the increase in the fiber size (hypertrophy) that coincides with the decrease in the number of MN in these animals (Table 3).

### 2.4. PCR Array

To gain insight into changes in skeletal muscle genes and the pathways they regulate as a result of HFD feeding, we conducted PCR analyses on the TA muscles from control and HFD animals of both sexes. Our results revealed that of the 84 key genes associated with skeletal muscle differentiation, function, and disease, 43 genes in males and 54 genes in females were significantly altered by an HFD (*p* < 0.05). Of these, 27 genes were upregulated and 16 were downregulated in the HFD male muscle samples (Figure 7A); however, 48 genes were upregulated, and six were downregulated in the HFD female muscle samples (Figure 7B).

The significantly and differentially expressed genes were grouped according to their functional significance using gene ontology terms. These genes were classified into 14 functional categories in males: signal transduction, positive regulation of muscle development, positive development of myoblast differentiation, negative development of fat cell differentiation, negative regulation of the MAPK cascade, cellular response to insulin stimulus, cell differentiation, muscle filament sliding, regulation of glucose metabolic process, insulin receptor signaling pathway, skeletal muscle development, muscle contraction, skeletal muscle contraction, and cellular glucose homeostasis (Figure 7C). These genes were classified into 15 functional categories in females: positive regulation of vascular smooth muscle cell proliferation, cellular response to fatty acids, cellular response to cadmium ions, positive regulation of muscle cell differentiation, regulation of DNA-binding transcription factor activity, muscle cell cellular homeostasis, positive regulation of glucose import, positive regulation of neuron differentiation, positive regulation of myoblast differentiation, cellular response to insulin stimulus, muscle filament sliding, regulation of glucose metabolic process, skeletal muscle tissue development, muscle contraction, and skeletal muscle contraction (Figure 7D).

### 2.5. Quantitative Real-Time PCR 

To validate the PCR array expression results and to verify the changes in SC distribution and the frequency of the muscle fiber types (IIA, IIX, and IIB), we performed RT-qPCR experiments on the key fiber-specific genes: myosin heavy chain 2 (*Myh-2, MyHC-IIa*), myosin heavy chain 1 (*Myh-1, MyHC-IIx*), myosin heavy chain 4 (*Myh-4, MyHC-IIb*), satellite cell-specific gene *Pax7*, and the muscle size regulator gene myostatin (*Mstn*). We used the same RNA samples from the TA muscles utilized for the PCR array analyses. *Gapdh* expression was used as the normalization standard, which was consistent in all the samples and the groups analysed. The qRT-PCR results confirmed that expression changes were comparable to those obtained from the PCR array (Figure 8). This suggested that the genes identified from the array were genuine targets. These results also confirmed the changes obtained with the fiber morphology analyses (Table 1 and Table 2).

## 3. Discussion

This study demonstrates the effects of sustained HFD feeding on skeletal muscle fibers at both cellular and molecular levels. Our results show that these effects do not differ based on sex since both male and female animals showed similar trends. There was a significant decrease in the frequency of oxidative fibers in both sexes owing to the sustained HFD feeding. BODIPY staining revealed prominent intramyocellular lipid accumulation in the forms of lipid microdroplets and sub-sarcolemmal lipid plaques. At the cellular level, the study revealed that long-term feeding with an HFD induced significant decreases in the numbers of SCs and MN within the muscle fibers, which indicates a reduction in muscle repair and regenerative capability. This was further confirmed by molecular studies, which showed a decrease in the expression level of the *Pax7* gene and an increase in the expression of the *Mstn* gene.

Feeding with an HFD induced a significant increase in the body weights of both male and female mice after eight and five weeks, respectively. The three weeks difference between males and females could be attributed to sexual differences in hormonal and metabolic activities. Despite the increase in body weight, there was a significant decrease in the weight of the TA muscle in HFD animals compared to that of control animals. This is comparable to a previous study on the short-term (six weeks) effects of HFD, where the TA muscle of HFD mice had decreased mass compared to that of control mice [15]. However, other studies reported no difference in mouse TA muscles after HFD feeding [10,14]. Despite reduction in muscle weight, the size of the oxidative muscle fibers (IIA and IIX) increased significantly. We speculate that this is a non-functional increase owing to the accumulation of intramyocellular lipids. BODIPY staining of HFD muscles showed a prominent accumulation of intramyocellular lipids in two forms: intracellular microdroplets and sub-sarcolemmal plaques. Additionally, a recent study reported that lipid accumulation in HFD-fed mice was most prominent in type IIA and IIX fibers [18].

Sustained HFD feeding induced a significant decrease in the frequency of oxidative fibers in both male (IIA and IIX) and female (IIA) mice. This could be explained by either transformation into a glycolytic fiber type or degeneration of oxidative fibers within the muscle. We are in favor of the latter hypothesis since it is supported by two important findings. First, there was no significant increase in the frequency of glycolytic (IIB) fibers within the muscles of HFD animals. Second, the diminution in muscle weight of HFD animals that supports the hypothesis of muscular atrophy and maybe fiber loss. However, further investigations are warranted to examine the hypothesis of oxidative fiber degeneration after an HFD feeding.

Several previous studies have reported the effects of obesity on skeletal muscle fiber type. Thomas et al. reported an early oxidative shift in mouse TA muscle fibers after three weeks of HFD feeding [14]. However, Shortreed et al. reported no changes in fiber type percentages within mouse TA muscles after eight weeks of HFD feeding [10]. Similarly, Hua et al. demonstrated no significant fiber type shifting in mouse TA muscles after six weeks of HFD feeding [15]. However, they found a significant decrease in type I fibers and an increase in type II fibers within the lumbar muscles of the same experimental animals [15]. In humans, several studies support the presence of a negative association between adiposity and the percentage of type I oxidative fibers [9,16,17]. For example, body fat percentage was negatively correlated with the proportion of type I oxidative fibers and positively correlated with the proportion of type IIB glycolytic fibers in Pima Indians [17]. Our findings agree with most of the literature suggesting a negative association between obesity and oxidative type skeletal muscle fibers.

Satellite cells are the primary contributors to skeletal muscle growth, repair, and regeneration processes [3,6,19]. Damaging stimuli to these cells can eventually lead to prominent muscle atrophy and impaired locomotor function [20]. Previous studies have shown that HFD-induced obesity is associated with a decreased number and function of SCs [13,20,21]. D’Souza et al. anticipated that an HFD attenuates skeletal muscle repair and regenerative capacity by impairing SC functionality [22]. However, they did not observe a decline in SC number in the HFD group, but rather an inability to activate and proliferate after an eight-week HFD feeding period. It is well known that SCs maintain their numbers through proliferation, as some proliferated SCs do not differentiate into MN; instead, they return to quiescence to conserve the SC pool [23]. Therefore, the inability of SCs to proliferate will eventually lead to loss of the SC reservoir, which was revealed in our study after 20 weeks of HFD feeding.

Previous studies have shown that HFD impairs regenerative muscle capacity after injury [24,25,26,27]. For example, delayed regeneration process, increased fibrosis, and decreased SC numbers were observed after a blunt injury to the hindlimb muscles of obese mice compared to normal-weight mice. Additionally, SC-related genes were also impaired in obese mice after trauma [26]. Geiger et al. found that HFD impairs SC proliferation, differentiation, and self-renewal in vitro. They also showed that HFD compromises SC properties in mice TA muscles after cardiotoxin injury [27].

In this study, to verify the changes observed in morphological and cellular studies, we analyzed the expression changes in 84 genes, which are known to be involved in skeletal muscle contractility, myogenesis, hypertrophy, and autocrine signaling, as well as genes involved in metabolic syndrome and wasting conditions. Our results show that 14 functional groups changed in males and 15 in females owing to HFD. These functional groups were approximately similar between both sexes. 

One of the genes whose expression was increased in HFD animals was *Mstn*, an essential regulator of muscle development and differentiation. Mstn protein belongs to the transforming growth factor-β superfamily, and earlier studies showed that it negatively regulates skeletal muscle development by inhibiting cell cycle progression and thus myoblast proliferation [28,29]. Furthermore, the absence of Mstn increases muscle mass in different animal species through hyperplasia and hypertrophy [30]. Interestingly, decreasing *Mstn* expression has been suggested as a therapeutic target for obesity [31]. 

Other key genes whose expressions are critical for myofiber type and contraction are the *Myhs*. Different *Myhs* are present in different species, and each myofiber type expresses different isoforms of Myosin. *Myh7* (*MyHC-I*) is predominantly expressed in slow-type myofibers, whereas adult fast-type myofibers express *Myh1* (*MyHC-IIx*), *Myh2* (*MyHC-IIa*), *Myh4* (*MyHC-IIb*), or *Myh13* (*MyHC-Eo*). Other isoforms are also expressed during embryonic development [32]. We observed decreased expression of *Myh1* and *Myh2* (predominantly expressed in fast oxidative myofibers), which confirms the reduction in oxidative myofibers.

Pax7 is a well-established transcription factor required to establish the muscle precursor cells during early embryonic developmental stages [4,23]. As mentioned earlier, Pax7 is specifically expressed in myogenic SCs [3,4,6,19,23], and previous studies have shown that it regulates muscle-specific transcriptional factors, especially through its coordination with the myogenic determination factor 1 (MyoD). A Pax7+/MyoD- SC indicates a quiescent SC, whereas Pax7+/MyoD+ cells are activated SCs. Once differentiated into a new myonucleus, the SC nucleus will cease expression of Pax7 [23]. The fact that *Pax7* expression was downregulated in both sexes further supports the decrease in SC numbers observed in our cellular studies.

One of the limitations to this study is that we examined only the TA muscle. Examining additional muscles with different fiber type compositions (i.e., soleus) may result in more informative data. In addition, some researchers may argue that the study included a low number of animals (*n* = 3/group), and adding more animals is recommended to infer more powerful statistical conclusions. However, the statistical analyses revealed that the data of this study were normally distributed, and there were strong correlations between the sample size and the results (observed power = 0.999, *p* < 0.001). Another important limitation is the lack of western blotting analysis. Western blotting analysis can complement this study’s investigations and add further evidence to its conclusions since the transcription level of a specific gene does not always reflect on its translational level. A final limitation was the lack of identifying hybrid fibers. Within the skeletal muscle, a number of fibers can co-express two or more different isoforms of myosin heavy chain. These are known as “hybrid fibers” [33]. Unfortunately, we could not compare the frequency of these fibers between the control and HFD animals.

In conclusion, long-term feeding of an HFD induces atrophy in the TA muscle, identified by muscle weight reduction, and decreases in all SC and myonuclear parameters. This atrophy was associated with a decrease in the percentage of oxidative fibers within the muscle. These findings were further confirmed by molecular experiments, which showed significant reductions in the expression of *Pax7*, *Myh1*, and *Myh2* genes. Male and female animals showed similar trends in response to HFD-induced obesity. Together, our findings indicate that the long-term effects of obesity on skeletal muscle resemble those of age-related sarcopenia.

## 4. Materials and Methods

### 4.1. Experimental Model

Adult male and female mice of the C57BL/6 strain were used in the present study. All animal care procedures and treatments were approved by the Animal Research Ethics Committee of the UAE University (approval ID: ERA_2019_5847). At eight weeks of age, male and female mice were randomly divided into two groups each (*n* = 3 mice/group): male and female control (C) groups that were fed a normal chow diet, and male and female HFD groups that were fed an HFD. The HFD was purchased from the Research Diet Incorporation (Cat. # D12492, Research Diet Inc., New Brunswick, NJ, USA). This HFD corresponds to the normal chow diet components, with the exception that 60% of kcal is from fat [34,35,36]. The detailed formulation of this HFD is available through the company website (https://researchdiets.com/formulas/d12492, accessed on 19 August 2022).

The mice were housed in large cages (3 mice/cage) at a room temperature of 21 °C and fed ad libitum for 20 weeks. Food and drinking water were available without restricted access. A light/dark cycle of 12/12 h was maintained. The animals were weighed on a weekly basis at the end of each week. At the end of 20 weeks, the mice were weighed and euthanized via inhalation of diethyl ether. The tibialis anterior (TA) muscles were dissected from both sides and weighed. The mouse TA muscle consists mostly of fast-twitch muscle fibers (types IIA, IIX, and IIB).

### 4.2. Tissue Preparation and Sectioning

The left TA muscles were used for cellular studies that included immunocytochemical investigations of fiber typing, intracellular lipids, and SC and myonuclear distribution. The muscles were coated with optimal cutting temperature (OCT) compound and fresh-frozen in 2-methylbutane cooled via liquid nitrogen. The muscles were then directly stored at −80 °C. The right TA muscle was used for molecular studies, including PCR arrays and qRT-PCR experiments. The muscles were directly frozen in liquid nitrogen and stored at −80 °C.

Serial cross-sections (10 μm-thick) were cut from the OCT-coated left TA muscles at 20 °C using a cryostat. Four continuous serial sections were picked up on each microscopic slide. Serial slides bearing sections were numbered and stored at −40 °C. Along with the cross-sections, longitudinal sections were also obtained to measure the lengths of the SC nuclei (SCN) and myonuclei (MN).

### 4.3. Fiber Typing Analysis

#### 4.3.1. Primary Antibodies

The following mouse monoclonal primary antibodies were used to determine the fiber type and size: (1) BF-F3 from the Developmental Studies Hybridoma Bank (DSHB; 1:50 dilution) to detect myosin heavy chain type IIb (fast-twitch glycolytic); (2) SC-71 (DSHB; 1:50 dilution) to detect myosin heavy chain type IIA (fast-twitch oxidative glycolytic); (3) 6H1 (DSHB; 1:50 dilution) to detect myosin heavy chain type IIX (fast-twitch glycolytic); (4) BA-F8 (DSHB; 1:50 dilution) to detect myosin heavy chain type I (slow-twitch oxidative) [37]. (5) We also used a rabbit polyclonal anti-laminin primary antibody (L9393; Sigma-Aldrich, St. Louis, MO, USA) at a 1:400 dilution to detect the basal laminae of skeletal muscle fibers [38].

#### 4.3.2. Immunocytochemical Protocol

The immunocytochemical experiments for fiber type were performed on four different sets; each set was selected for a specific myosin heavy chain (fiber type) antibody. Microscopic slides bearing sections were removed from −40 °C storage and air-dried for 15 min. A blocking solution consisting of 5% bovine serum albumin (BSA) in 1× phosphate buffer saline (PBS) was applied over the sections for 30 min at room temperature. The blocking solution was then drained from each slide. A primary antibody cocktail, containing two primary antibodies, anti-laminin and one of the aforementioned fiber-type antibodies, diluted in blocking solution, was applied over the sections. The sections were then incubated with primary antibodies for 48 h at 4 °C. After 48 h, the slides were washed three times with a blocking solution for 10 min each, and secondary antibodies diluted in the blocking solution were applied over the sections on each slide for 2 h at room temperature. Alexa Fluor^®^ 594-conjugated (diluted 1:200) and Alexa Fluor^®^ 488-conjugated (diluted 1:400) secondary antibodies were used to reveal the fiber type in red and basal lamina in green, respectively, when viewed with an epifluorescent microscope. After 2 h, the slides were washed thrice in blocking solution for 10 min each, and followed by two washes in PBS (1×) solution for 5 min each. Finally, an aqueous mounting medium (Thermo Scientific™ Shandon™ Immu-Mount™, Thermo Fisher Scientific, Waltham, MA, USA, #9990402) was applied with coverslips over the sections.

#### 4.3.3. Image Analysis and Data Collection

Images from the immunofluorescence-labeled sections were captured at 200× magnification using a BX43 LED Olympus microscope equipped with epifluorescence and DP74 Olympus digital still camera (Olympus Corporation, Tokyo, Japan). Ten different fields of view were obtained from the sections on each slide. Four fields were captured from the corners of each section. The capturing started at a different corner on each section and continued in a clockwise manner. Thus, for the first section, the first field captured was in the top-left corner of the section, whereas the second field was captured from the top-right corner of the same section. In the second section, the capturing continued with field five on the top-left corner of the section, etc. From each field, two fluorescent images, each viewed through a different wavelength filter, were acquired. The resultant images were labeled with myosin heavy chain type in red and basal laminae in green. The images were subsequently transferred to a desktop computer and superimposed using the Adobe Photoshop software (Adobe Inc., San Jose, CA, USA). Around 700 contiguous fibers were numbered from the fields of view, and the frequency of positively labeled fibers of these 700 fibers was calculated. 

Additionally, the ellipse minor axis, which represents the lesser fiber diameter, was measured to assess the size of the positively labeled fibers. The minor axis was used to surmount the distortion in the muscle fiber cross-section if it was cut at an angle other than transversely. The basal lamina images were used to measure the diameter of individual fibers using ImageJ software (an image analysis program developed by the National Institutes of Health and freely available for download (https://imagej.nih.gov/ij/, accessed 19 August 2022).

### 4.4. Labeling of Intracellular Lipids

#### 4.4.1. Labeling Agent

BODIPY™ 493/503 (4,4-Difluoro-1,3,5,7,8-Pentamethyl-4-Bora-3a,4a-Diaza-s-Indacene, C14H17BF2N2) was used at a concentration of 0.5 mg/mL in dimethyl sulfoxide (DMSO) to stain the intracellular lipids of the skeletal muscle fibers (cat. # D3922, Invitrogen by Thermo Fisher Scientific, Waltham, MA, USA) [39].

#### 4.4.2. Staining Protocol

Microscopic slides holding sections were removed from −40 °C storage, air-dried for 10 min, fixed with 4% formaldehyde in cold PBS (1×) for 10 min, and washed twice in fresh PBS for 5 min each. The slides were then incubated in the BODIPY solution for 30 min at room temperature. The slides were then rinsed twice for 5 min each in PBS (1×). Finally, Shandon™ Immu-Mount™ aqueous mounting medium (Thermo Fisher Scientific) was applied with coverslips over the sections.

### 4.5. Satellite Cell and Myonuclear Distribution

#### 4.5.1. Primary Antibodies and Nuclear Labeling Agent

The following primary antibodies and the nuclear labeling dye were used to identify SCN and MN: (1) Anti-Pax7 (DSHB), a mouse monoclonal antibody (diluted 1:10) to label SC nuclei [3,19]; (2) anti-laminin (Sigma-Aldrich, St. Louis, MO, USA, #L9393), a rabbit polyclonal antibody (diluted 1:400) to detect the basal laminae of skeletal muscle fibers [38]; and (3) DAPI (4′,6-diamidino-2-phenylindole), a nuclear counterstain that emits blue fluorescence upon binding to the AT regions of the DNA.

#### 4.5.2. Immunocytochemical Protocol

Microscopic slides holding sections were removed from −40 °C storage, air-dried for 10 min, fixed with 4% formaldehyde in cold PBS (1×) for 10 min, and washed twice in fresh PBS for 5 min each. Thereafter, the slides were incubated in ice-cold methanol for 15 min for permeabilization, followed by three 5 min washes with PBS (1×). The slides were then transferred to a slide container filled with 200 mL citrate buffer (10 mM sodium citrate, 0.05% Tween 20, pH 6.0) for antigen retrieval. The container with the slides was then transferred to a microwave oven and boiled for 2 min at high power (1000 W), followed directly by boiling for 20 min at low power (200 W). The container was cooled to room temperature for 30 min, followed by three washes in PBS (1×) for 5 min each.

After that, the slides were incubated in mouse-on-mouse blocking reagent (MKB-2213-1, Vector Laboratories) for 45 min at room temperature to block endogenous mouse immunoglobulins. The slides were then rinsed three times for 5 min each in PBS (1×). Another blocking solution, composed of 10% BSA in PBS (1×) was used to block the sections on the slides for an additional 30 min at room temperature. The blocking solution was then drained from each slide and anti-Pax7 and anti-laminin primary antibodies diluted in the BSA blocking solution were applied over the sections on the slides. The slides were incubated overnight with the primary antibodies at 4 °C. 

The following day, slides were washed thrice in fresh PBS (1×) for 5 min each, and biotinylated goat anti-mouse IgG (H+L) antibody (cat # B2763, Thermo Fisher Scientific) diluted 1:500 in PBS (1×) was applied over the sections for 1 h at room temperature. The slides were then washed three times in fresh PBS (1×) for 5 min each. Streptavidin-horseradish peroxidase conjugate (cat. # S911, Thermo Fisher Scientific) diluted 1:500 in PBS (1×), and Alexa Fluor^®^ 488-conjugated goat anti-rabbit secondary antibody (cat # A11034, Thermo Fisher Scientific) diluted 1:400 in PBS (1×) solution were applied together over the sections for 1 h at room temperature. The slides were then washed thrice in fresh PBS (1×) for 5 min each. Tyramide signal amplification reagent conjugated with Alexa Fluor™ 594 fluoro-dye (cat # B40957, Thermo Fisher Scientific) was diluted 1:200 in PBS (1×) and applied over the sections for 20 min at room temperature to reveal SCN in red under an epifluorescence microscope. The slides were then washed thrice in fresh PBS (1×) for 5 min each. Finally, an aqueous mounting medium containing DAPI (Fluorshield, Abcam, Cambridge, UK, #ab104139) was applied with coverslips over the sections.

#### 4.5.3. Image Analysis and Data Collection

Images from the immunofluorescence-labeled sections were captured at 200× magnification using a BX43 LED Olympus microscope equipped with epifluorescence and a DP74 Olympus digital still camera (Olympus Corporation). Five different fields of view were obtained from the sections on each slide following the same protocol mentioned above in fiber typing analysis. From each field, three fluorescent images, each viewed through a different wavelength filter, were acquired. The resultant images were labeled with all nuclei in blue, SCN in red, and basal laminae in green. The images were subsequently transferred to a desktop computer and superimposed using the Adobe Photoshop software. 

The numbers of SCN and MN per fiber cross-sectional profile were counted for 200 contiguous fibers from each muscle. The frequency of SCs was calculated for each animal according to the following equation: Frequency = [SCN/(SCN + MN)] × 100%. 

In addition, the numbers of SCs and myonuclei per unit length of fiber (1 mm) were calculated using the equation: Number = A/(Ln + M), where A is the mean number of nuclei per fiber cross-sectional profile, Ln is the mean nuclear length, and M is the thickness of the tissue section [3,4,5,6]. Longitudinal sections were used to measure SCN and MN lengths. The same immunocytochemical protocol described above was used to study the longitudinal sections. For each animal, the lengths of 10 SCN (30/group) and 40 MN (120/group) were measured using ImageJ software. 

The SC concentration was measured as the surface area of the muscle cell membrane (sarcolemma) per each SC. First, the surface area of the sarcolemma per unit length of fiber was calculated for each animal using the equation: Area = πEU, where E is the ellipse minor axis (fiber diameter) and U is the unit length of the fiber (1 mm). The area of the sarcolemma per SC was determined by dividing the calculated surface area by the number of SCs per mm in each animal. Finally, the size of the myonuclear domain (MD) was determined by dividing the volume of sarcoplasm per unit length of fiber by the number of MN in that unit length (1 mm). The volume of sarcoplasm per unit length was computed according to the equation: Volume = π(E/2)2 × U, where E is the ellipse minor axis (fiber diameter) and U is the unit length of the fiber (1 mm).

### 4.6. PCR Profiler Array for Myogenesis

#### 4.6.1. RNA Preparation and Reverse Transcription

Total RNA was isolated from mouse TA muscles using easy-BLUE (iNtRON Biotechnology, Seongnam, South Korea, #17061) according to the manufacturer’s recommendations. Total RNA was treated with RNase-free DNase I (Lucigen-LGC, Middleton, WI, USA, #D9905K) to remove genomic DNA contamination, and was purified using the MasterPure™ Complete DNA and RNA Purification Kit (Lucigen-LGC, #MC85200). The quantity and quality were assessed using a Nanodrop (Thermo Fisher Scientific). Five micrograms of RNA were subjected to first-strand cDNA synthesis using SuperScript™ IV Reverse Transcriptase (Invitrogen by Thermo Fisher Scientific, #18090010).

#### 4.6.2. PCR Array

A PCR array kit (QIAGEN, Hilden, Germany, #330231 PAMM-099ZA), containing 84 key genes associated with skeletal muscle differentiation, function, and disease was used. The PCR array experiments were carried out according to the manufacturer’s recommendations. Approximately 1 µg of cDNA was used for one 96-well array plate. The PCR array was performed based on SYBR Green chemistry in a QuantStudio™ 7 Flex Real-Time PCR system (Thermo Fisher Scientific). All PCR array experiments were performed in triplicate, and the relative expression was calculated using the delta–delta CT method, as previously described [40].

#### 4.6.3. Bioinformatics Analysis

Differentially expressed genes from the PCR array were used for a gene ontology (GO) study using pathfindR (R package) [41]. Enriched GO-terms were visualized as bubble charts using ggplot2 (R package). All significantly altered genes in the PCR array were visualized as heatmaps using the ComplexHeatmap (R package) [42].

### 4.7. Real-Time Quantitative PCR

Genes that were significantly altered in the array and were associated with skeletal muscle fiber type and muscle function were quantified using RT-qPCR. The expression of these genes was analyzed using the delta–delta CT method. The primers used in the RT-qPCR are listed below.
*Pax7*
Forward primerCAGCAAGCCCAGACAGGTGReverse primerCCGGATTTCCCAGCTGAACA*Myh1*
Forward primerCACGCTGGATGCTGAGATTAReverse primerAGGTGCAGCTGAGTGTCCTT*Myh2*
Forward primerGAATGCCTACGAGGAGTCTCTReverse primerTTCTGCAATCTGTTCCGTGA *Myh4*
Forward primerCAAGTCATCGGTGTTTGTGGReverse primerGGCCATGTCCTCAATCTTGT*Mstn*
Forward primerCCCCCTCACGGTCGATTTTGReverse primerGGTGCACAAGATGAGTATGCG*Gapdh*
Forward primerGGTGCTGAGTATGTCGTGGAGReverse primerTTCTCGTGGTTCACACCCAT

### 4.8. Statistical Analysis

The data were sampled based on sex. Within each sex, the data for the measured parameters were blocked into two groups: control vs. experimental (HFD). All numerical data were expressed as mean ± standard deviation (SD). Levene’s test for equality of variances was first applied to determine the homogeneity of variance between control and experimental groups. Data were then evaluated for each sex by independent samples *t*-tests at the 5% level of significance. Statistical analyses were performed using IBM SPSS Statistics software (standard version 28.0.0.0, IBM, Armonk, NY, USA).

## 5. Conclusions

Long-term feeding of an HFD induces atrophy in the TA muscle, identified by muscle weight reduction, and decreases in all SC and myonuclear parameters. This atrophy was associated with a decrease in the percentage of oxidative fibers within the muscle. These findings were further confirmed by molecular experiments, which showed significant reductions in the expression of Pax7, Myh1, and Myh2 genes. Male and female animals showed similar trends in response to HFD-induced obesity. Together, our findings indicate that the long-term effects of obesity on skeletal muscle resemble those of age-related sarcopenia.

## Figures and Tables

**Figure 1 ijms-23-09547-f001:**
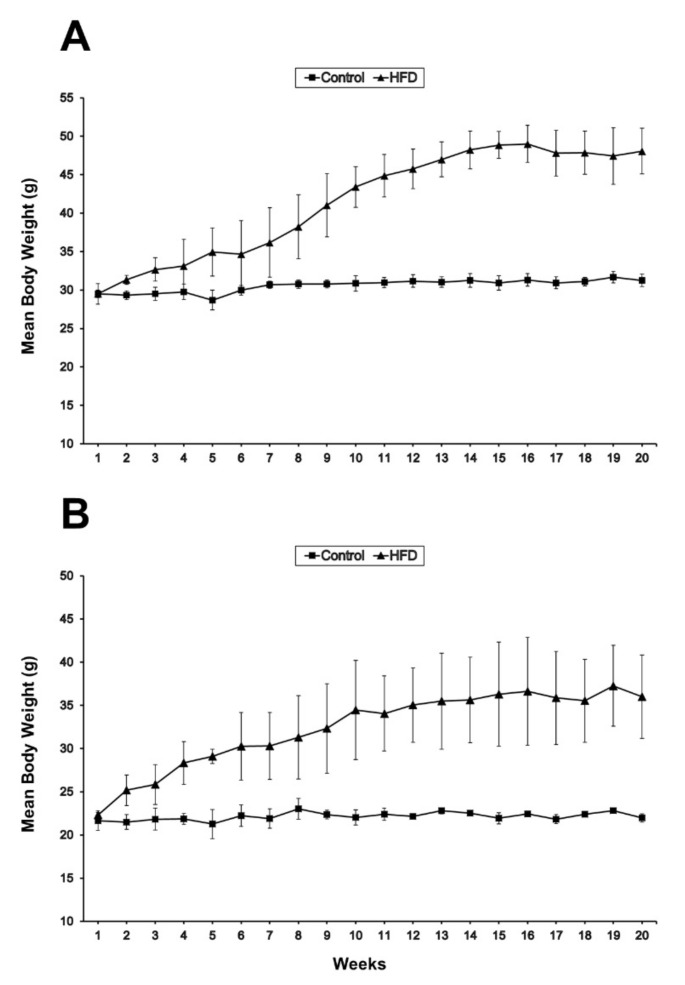
Variations in body weight with time between standard chow (control) and high-fat diet (HFD)-fed mice of both sexes (*n* = 3 mice/group). (**A**) In males, the increase in body weight due to HFD feeding becomes significant (*p* < 0.05) by the eighth week. (**B**) In females, the increase becomes significant (*p* < 0.05) by the fourth week.

**Figure 2 ijms-23-09547-f002:**
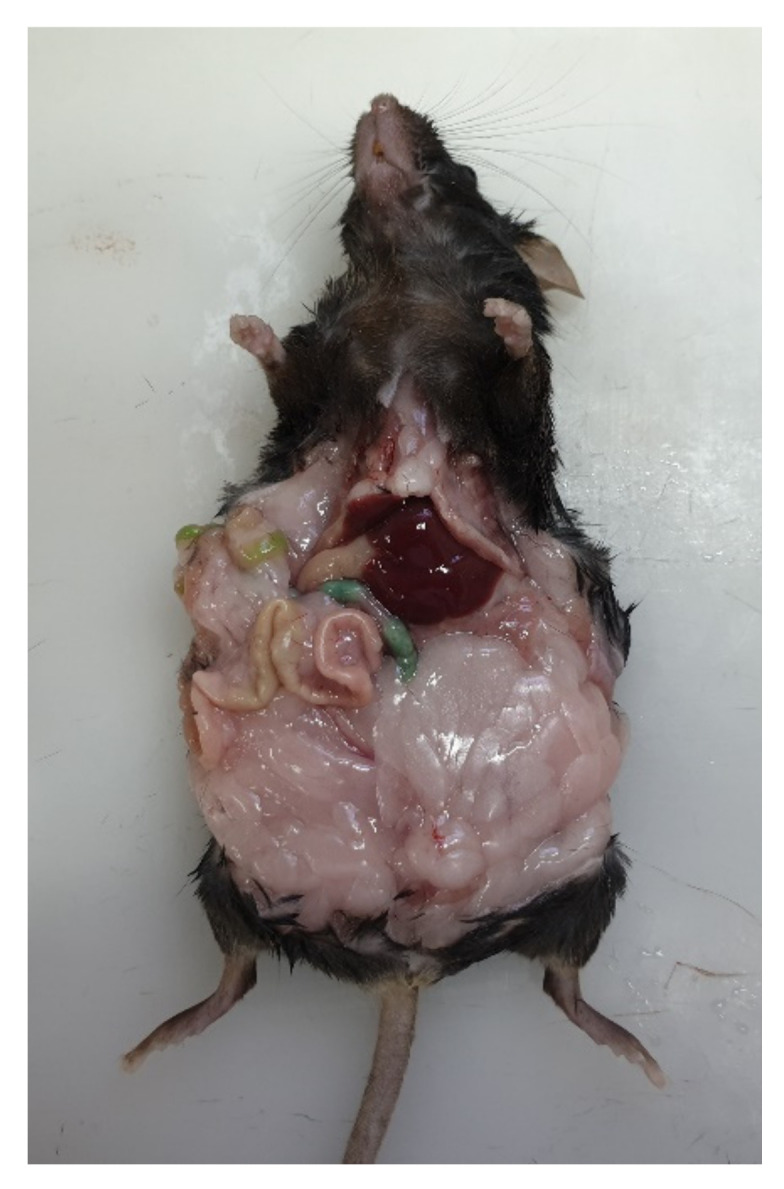
A high-fat diet-fed male mouse showing a prominent accumulation of visceral fat.

**Figure 3 ijms-23-09547-f003:**
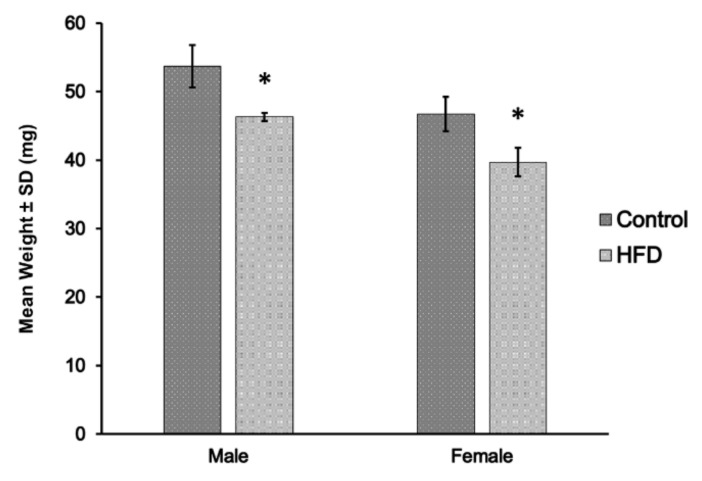
Tibialis anterior muscle weight between standard chow (control) and high-fat diet (HFD) groups of male and female mice (*n* = 3 mice/group). There was a significant (* *p* < 0.05) decrease in muscle weight in HFD groups compared to corresponding control groups in both sexes.

**Figure 4 ijms-23-09547-f004:**
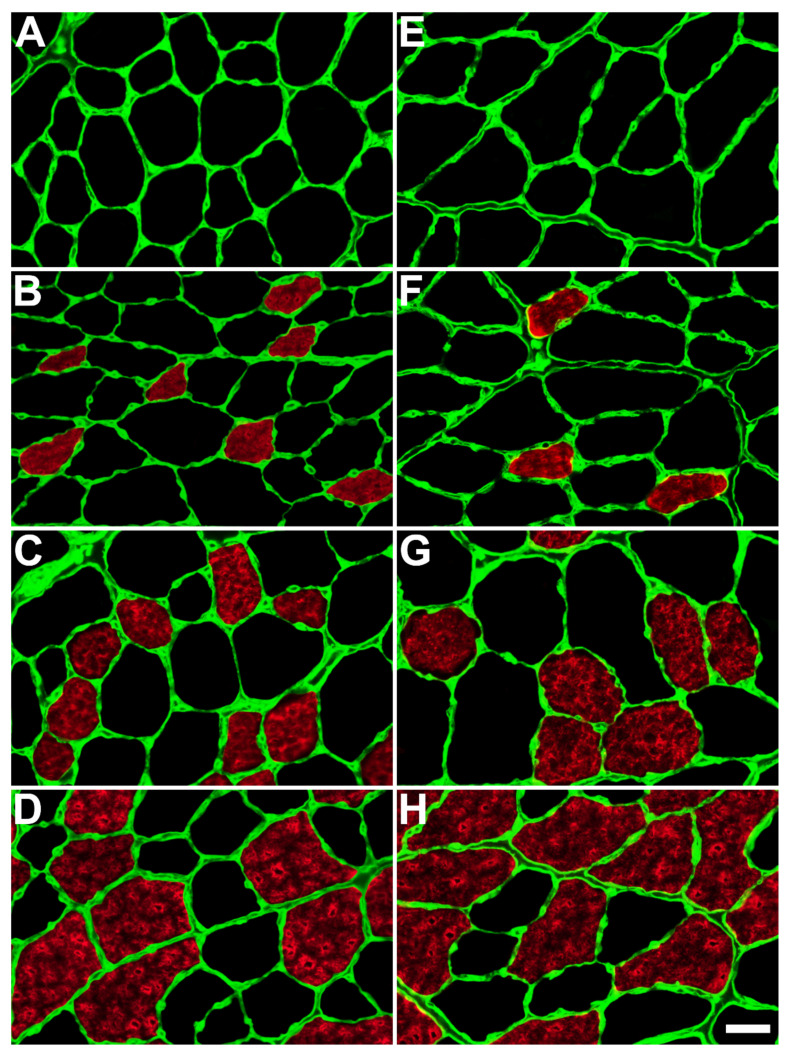
Immunocytochemical labeling for different fiber types within the tibialis anterior muscle of control (**A**–**D**) and high-fat diet-fed (**E**–**H**) male mice. The figure shows immunolabeling for type I fibers (**A**,**E**), immunolabeling for type IIA fibers (**B**,**F**), immunolabeling for type IIX fibers (**C**,**G**), and immunolabeling for type IIB fibers (**D**,**H**). There were no type I slow fibers, whereas the most frequent fibers were type IIB. Scale bar = 30 μm.

**Figure 5 ijms-23-09547-f005:**
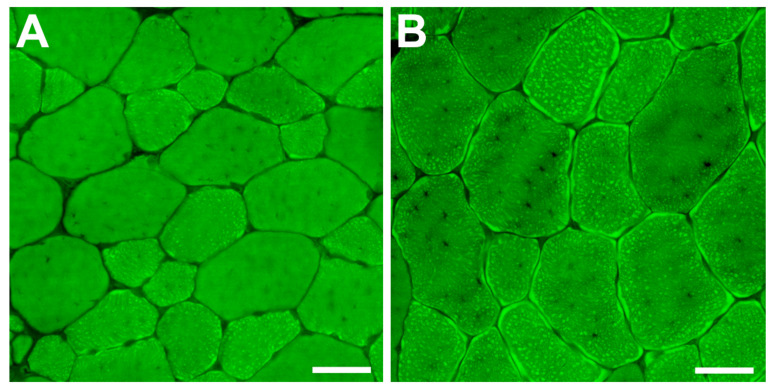
BODIPY staining for intramyocellular lipids within the tibialis anterior muscle fibers. (**A**) Represents a section from a standard chow animal. (**B**) Represents a section from a high-fat diet (HFD) animal. There was a prominent accumulation of intramyocellular lipids in the HFD animal in the forms of microdroplets and sub-sarcolemmal plaques. Scale bar = 30 μm.

**Figure 6 ijms-23-09547-f006:**
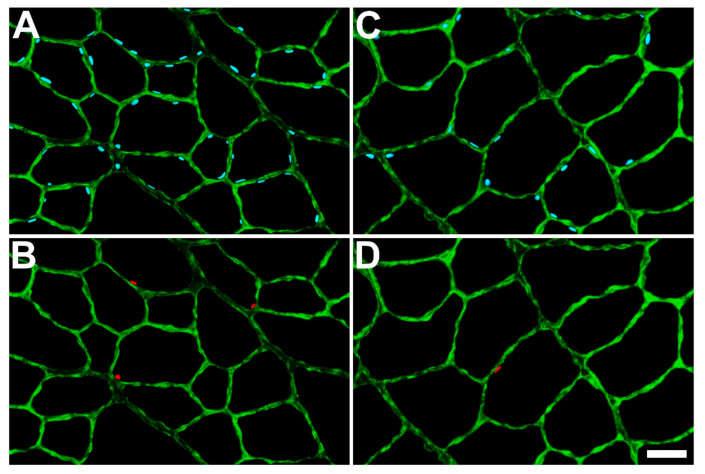
Immunocytochemical identification of satellite cell nuclei (SCN) and myonuclei (MN) in cross-sections of tibialis anterior muscles obtained from standard chow (control) and high-fat diet (HFD)-fed mice. (**A**,**B**) represent a section obtained from a control mouse. (**C**,**D**) represent a section obtained from an HFD mouse. (**A**,**C**) show all nuclei in blue (DAPI staining) and the basal laminae of muscle fibers in green (anti-laminin staining). (**B**,**D**) show SCN in red (anti-Pax7 labeling) and the basal laminae in green. Scale bar = 30 μm.

**Figure 7 ijms-23-09547-f007:**
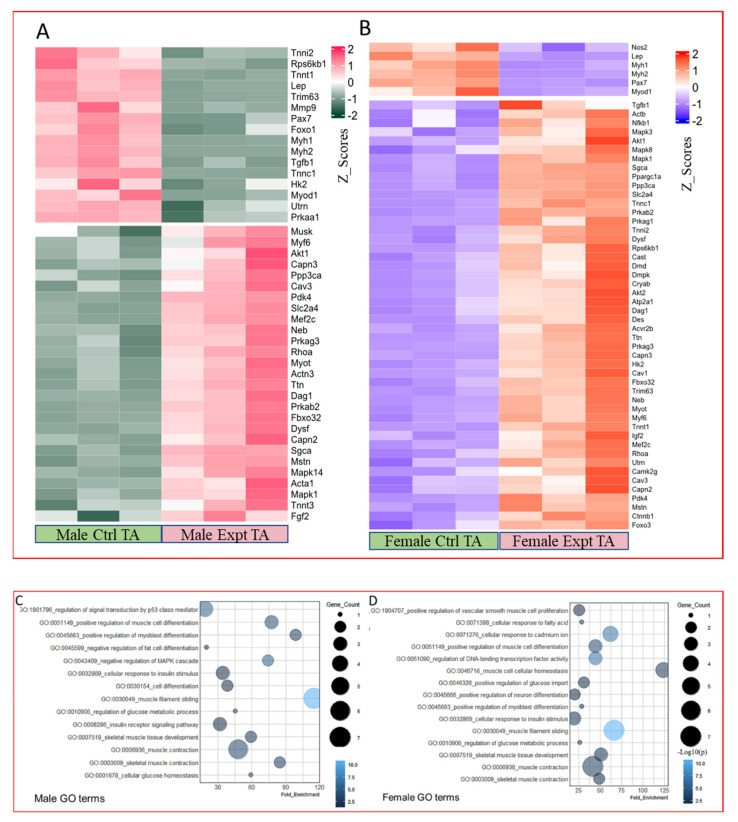
PCR array expression of genes associated with skeletal muscle differentiation, function, and disease between control (Ctrl) and high-fat diet (HFD) mice. (**A**) Male and (**B**) female heatmaps from tibialis anterior muscles of Ctrl and HFD animals showing the altered genes. (**C**) Male and (**D**) female GO terms of the muscle-specific pathways that are differentially affected between Ctrl and HFD animals. As illustrated, there were three mice per group.

**Figure 8 ijms-23-09547-f008:**
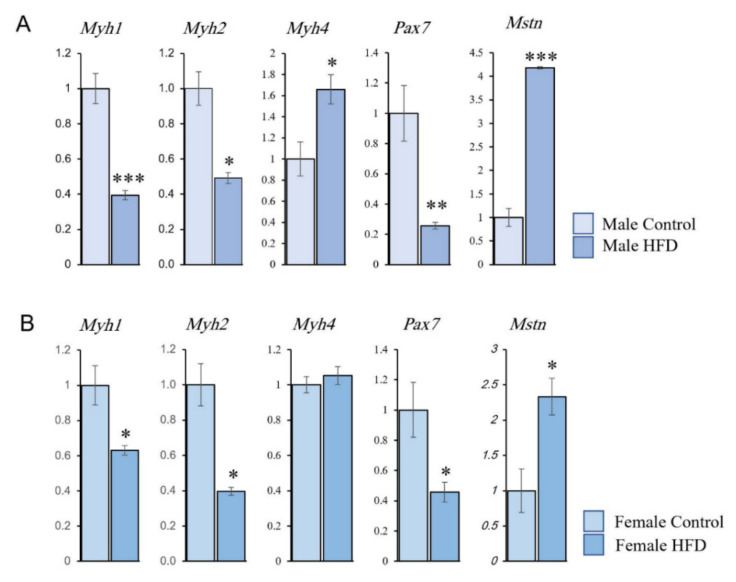
RT-qPCR of (**A**) male and (**B**) female tibialis anterior muscle revealed a reduction in the expression of muscle fiber-specific genes *Myh1* (IIX) and *Myh2* (IIA) in both sexes. An increase in *Myh4* (IIB) was seen in males, while there was no significant increase in females. *Pax7* expression was reduced in both males and females, while *Mstn* expression was increased in both sexes. * *p* < 0.05, ** *p* < 0.01, *** *p* < 0.001 significantly different from its corresponding control group (*t*-test).

**Table 1 ijms-23-09547-t001:** Variations in fiber type and size between control and high-fat diet (HFD)-fed male mice.

Fiber Type	Fiber Frequency (%)		Fiber Diameter (μm)	
	Control	HFD	Control	HFD
I	0.0	0.0	-	-
IIA	10.9 ± 1.0	7.4 ± 2.0 *	22.9 ± 1.9	30.8 ± 1.4 *
IIX	37.8 ± 1.5	32.7 ± 2.3 *	29.3 ± 1.0	35.5 ± 3.3 *
IIB	50.2 ± 4.3	59.5 ± 5.9	41.8 ± 0.5	41.6 ± 1.0

Abbreviations: %, percent; µm, micrometer; HFD, high-fat diet. The values are presented as the mean ± SD for three animals. From each animal, 700 contiguous fibers were counted for each fiber type. * (*p* < 0.05) significantly different from the corresponding control group.

**Table 2 ijms-23-09547-t002:** Variations in fiber type and size between control and high-fat diet (HFD)-fed female mice.

Fiber Type	Fiber Frequency (%)		Fiber Diameter (μm)	
	Control	HFD	Control	HFD
I	0.0	0.0	-	-
IIA	15.0 ± 1.3	8.6 ± 2.8 *	24.7 ± 2.5	24.9 ± 3.0
IIX	40.7 ± 8.1	32.8 ± 7.6	27.2 ± 1.0	32.8 ± 2.6 *
IIB	44.4 ± 6.4	57.8 ± 13.8	38.6 ± 3.2	39.0 ± 5.1

Abbreviations: %, percent; µm, micrometer; HFD, high-fat diet. The values are presented as the mean ± SD for three animals. From each animal, 700 contiguous fibers were counted for each fiber type. * (*p* < 0.05) significantly different from the corresponding control group.

**Table 3 ijms-23-09547-t003:** Distribution of satellite cells (SCs) and myonuclei (MN) in control and high-fat diet (HFD) groups of both male and female murine skeletal muscle fibers.

Sex	Male		Female	
Group	Control	HFD	Control	HFD
Fiber Diameter ^#^ (μm)	36.3 ± 0.7	41.1 ± 2.5 *	34.1 ± 1.7	38.0 ± 1.1 *
Length of SCN (μm)	9.9 ± 0.1	9.5 ± 0.4	10.3 ± 0.2	9.9 ± 0.2
SC Frequency (%)	7.0 ± 0.2	3.7 ± 0.1 **	7.3 ± 0.2	3.6 ± 0.2 **
SCs/mm	4.9 ± 0.4	1.8 ± 0.1 **	4.7 ± 0.5	1.7 ± 0.2 **
Surface area/SC (×10^3^ μm^2^)	23.6 ± 2.3	73.1 ± 4.5 **	23.1 ± 2.6	70.6 ± 6.5 **
Length of MN (μm)	10.4 ± 0.1	10.1 ± 0.5	10.1 ± 0.4	10.2 ± 0.5
MN/mm	63.6 ± 4.9	44.7 ± 1.1 **	61.4 ± 4.0	42.9 ± 3.8 **
MD (×10^3^ μm^3^)	16.4 ± 1.9	29.8 ± 4.0 **	15.0 ± 2.0	26.5 ± 2.0 **

Abbreviations: SC, satellite cell; SCN, satellite cell nucleus; MN, myonuclei; MD, myonuclear domain. Each value represents the mean ± standard deviation (SD). ^#^ The fiber diameter values are the means for the 200 contiguous fibers/mouse that were used to count SCs and MN (*n* = 3 mice/group). * *p* < 0.05, ** *p* < 0.01 significantly different from its corresponding control group (*t*-test).

## Data Availability

The data presented in this study are available on request from the corresponding authors.

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
