# Peer review of "Cellular and Molecular Variations in Male and Female Murine Skeletal Muscle after Long-Term Feeding with a High-Fat Diet"

_ijms, 2022, doi:10.3390/ijms23179547_

Round 1
Reviewer 1 Report
Here the authors aimed to gain insight into the modulating effects of HFD on skeletal muscle function. There are a variety of interesting observations:
1. The authors gave data for the alteration of fiber type composition and lipid accumulation in mouse Tibialis anterior (TA) following the application of HFD.
2. They gave evidence about the decreased frequency of SCs assuming an impaired regenerative capacity of the skeletal muscle.
3. They presented novel results about the altered expression of several muscle function, development and metabolism related genes.
(Recommendations for the authors):
Comments:
1. It would have been a better approach to examine soleus and EDL muscles in parallel, since it is quite difficult to clearly discuss the influence of HFD on fiber composition and metabolic shift, (considering that oxidative fiber parameters changed, on the other hand glycolytic fibers remained unaffected)? It might have resulted in a comparable, more informative data.
2. Please explain the scientific reason or significance of investigating the length of SC nuclei and MN.
3. Table 1, 2: Please indicate how many fibers were counted.
4. Figure 8: If normalized to GAPDH, despite the altered expression of numerous metabolism related genes, was the GAPDH proved to be constant in all of these samples?
5. Please explain the term Myonuclear domain (MND).
6. Row 440: “Differentiate into MN” should be corrected, since during assymetric cell division one of the daughter cells becomes precursor myoblast and not MN.
7. Is there any data in the scientific literature about investigating the injury induced muscle regeneration following HFD?
8. Table 3: When contiguous fibers were analysed, alteration in fiber diameter suggests hypertrophy. But the main conclusion is atrophy in TA. It is a bit confusing, because the size of the dominant IIB fibers has not changed.
9. May apoptosis be responsible for the decreased satellite cell frequency? Can it be examined somehow? Probably the decreased proliferation is not the only reason of that observation.
Reviewer 2 Report
The authors studied the effect of HFD on tibialis anterior muscle in male and female mice. They found that in both males and females, the muscles in obese mice showed signs of atrophy and shift towards fast type myosin heavy chain isoforms. The paper is interesting and has merit, however I have several objections.
It is not clear how many animals were in each study group. Were only 3 animals used for each group? Please indicate the number clearly in the text and in the Figure and table captions. How was the sample size determined?
How were the immunofluorescent imgaes captured. Were the view fields captured randomly, systematically? In my experience, tibialis anterior in mice is very heterogenous muscle, having a portion that is composed of more fast type fibres, a portion composed mor of a slower type fibres and mixed portion in between. With inappropriate sampling and small number of animals there could have been a significant bias. Please describe how was the sampling of the view fields performed. Can you provide a few pictures of a whole muscle cross-section where MyHC are marked?
In line 167 the concentration of bodipy is missing.
Please describe how was morphometric analysis performed. Why was the perimeter of muscle fibres calculated and not measured for each muscle fibre?
Why was the ellipse minor axis used to calculate fibre diameter? It was shown that equivalent circular area diameter or average Feret diameter are the most robust diameters. E. g. DOI: 10.1111/jmi.12985
Please describe how was statistical analysis performed. How was the normality of data distribution tested? A two-way ANOVA should be used to test the differences between groups (with factors sex and diet).
In the result section, the exact P values should be reported.
Was the content of lipid droplets quantified or semiquantified? The authors should attempt to semiquantify the accumulation of lipid droplets in each fibre type and statistically test if there are any significant differences between groups and muscle fibre types.
Can you calculate also areal density of muscle fibre types (proportion of cross-sectional area of each fibre type in the muscle). This may better correlate with gene expression changes and show the effect of each fibre type on muscle atrophy.
Line 403 replace “was lighter” with had “decreased mass”.
You found that the mass of the muscle was significantly lower in obese mice (for about 10-20 %) and that on average the diameter of muscle fibres increased. How would you explain these at first hand inconsistent results? Were the number of muscle fibres in whole TA muscle in obese groups lower than in control group? You favour the concept of muscular atrophy and fiber loss. This hypothesis of fibre loss cannot be speculated from your result since there are no controls from the beginning of the HFD at 8 weeks of age. Please rephrase this conclusion to be less speculative. Also did you found any morphological signs of atrophy or necrosis (centralisation of nuclei, fibre type grouping, increased heterogeneity of fibre diameter etc.)?
Were the hybrid fibres also analysed?
Line 478/479. Both male and female animals showed similar trends in response to the HFD-induced obesity. This conclusion should also be statistically tested for.
Reviewer 3 Report
Evaluation of the Manuscript ID: ijms-1753064
Dear Editor of the International Journal of Molecular Sciences, the paper of Emerald et al. “Cellular and Molecular Variation in Male and Female Murine Skeletal Muscle After Long-Term Feeding with High-Fat Diet” deals about the effects of long-term high-fat diet on skeletal muscle analysed at morphological, cellular and molecular levels. The main results reported in this paper showed a decrease of muscle (tibialis anterior) weight, the number of satellite cells and myonuclei from HFD mice consisting in an atrophic phenotype. This atrophy also was associated with a decrease in the percentage of oxidative fibers within the muscle, and a significant reduction in the expression of Pax7, Myh1 and Myh2 genes and an increase in the expression of Mstn gene. The manuscript is well written and organised. A great number of experiments and data are reported.
I do consider it an important contribution towards understanding the importance of high fat diet in sarcopenia.
I recommend publishing of this MS on International Journal of Molecular Sciences with a minor revision.
Since the antibody of Mstn, Pax7, Myh1 and Myh2 are commercially available, it would be beneficial for the reader to have a western blot analysing protein levels.
Reviewer 4 Report
Abstract: The concept of the paper suggesting a link between HFD-induced muscle atrophy and type II fibers, is weak. The fact that atrophy during HFD is associated with increased type II fibers may not be causal. During fasting, muscle atrophy is associated with increased type I fibers, and treatment with thyroid hormone in this situation prevents atrophy associated with increases the presence of type II fibers (PMID 31732814). Refs within 10-13 need to be re-addressed since fiber type changes are not always reported. Ref 10 reports no change in fiber type in tibialis anterior muscle. Which the authors study.
Introduction: line 63 needs references.
Major:
The number od animals for each experimental condition (n=3) is very low and easily allows for misinterpretations of the results. This study seems underpowered. To achieve the stated number of animals, did the authors perform a power test? The statistical analysis method, and associated post-hoc analysis in the methods section is lacking. Was the distribution of the obtained data normal?
Was the PCR array performed with pooled RNA from the 3 animals, or was it repeated 3 times? It seems that the sensitivity of the measurement in the female mice is higher, more genes are visualized. Which internal control gene was used? GAPDH, like in the PCR studies? When gender is compared, the analyses of male and female material should be performed within the same reaction in order to compare the expression of the different genes (Fig 7 and 8). This would change the statistical approach (ANOVA should be applied). Even when changes in expression are relatively the same, the absolute levels between genders mice may be much higher, influencing the basis for the metabolic phenotype. It is hard to interpret the heat map at-a-glance because the order of the genes is different, and in the males the decreased genes are plotted at the high-end whereas in the females at the low end.
In Fig.4, the control muscles are lacking, to underline the differences between the two conditions. At least 10 animals for each group are needed to observe statistically significant differences in fiber numbers, the surface needs to be indicated. It is also unclear why only males are shown, both genders need to be displayed. As this is a key point of the paper, these results, and their approproate statistical analysis, are mandatory to add to what is already known in the literature.
Minor: in Fig.2 please specify the gender
In the legend of each figure, please state the number of animals analyzed.
Round 2
Reviewer 2 Report
The authors addressed most of my comments satisfactorily. I have two minor additional comments.
In the limitation section the low number of animals should also be mentioned.
The strain of mouse used should be stated in abstract section.
Reviewer 4 Report
One more minor comment:
Line 170-173: indicate P-values (instead of 0.000, <0.0001)
